

# Improving parallel executions by increasing task granularity in task-based runtime systems using acyclic DAG clustering

Bérenger Bramas[1,2] and Alain Ketterlin[1,2,3]

[1] CAMUS, Inria Nancy - Grand Est, Nancy, France
[2] ICPS Team, ICube, Illkirch-Graffenstaden, France
[3] Université de Strasbourg, Strasbourg, France

## ABSTRACT

The task-based approach is a parallelization paradigm in which an algorithm is transformed into a direct acyclic graph of tasks: the vertices are computational elements extracted from the original algorithm and the edges are dependencies between those. During the execution, the management of the dependencies adds an overhead that can become significant when the computational cost of the tasks is low. A possibility to reduce the makespan is to aggregate the tasks to make them heavier, while having fewer of them, with the objective of mitigating the importance of the overhead. In this paper, we study an existing clustering/partitioning strategy to speed up the parallel execution of a task-based application. We provide two additional heuristics to this algorithm and perform an in-depth study on a large graph set. In addition, we propose a new model to estimate the execution duration and use it to choose the proper granularity. We show that this strategy allows speeding up a real numerical application by a factor of 7 on a multi-core system.

## INTRODUCTION

The task-based (TB) approach has become a popular method to parallelize scientific applications in the high-performance computing (HPC) community. Compared to the classical approaches, such as the fork-join and spawn-sync paradigms, it offers several advantages as it allows to describe the intrinsic parallelism of any algorithms and run parallel executions without global synchronizations. Behind the scenes, most of the runtime systems that manage the tasks use a direct acyclic graph where the nodes represent the tasks and the edges represent the dependencies. In this model, a task becomes ready when all its predecessors in the graph are completed, which causes the use a local synchronization mechanism inside the runtime system to manage the dependencies. There are now many task-based runtime systems (*Danalis et al., 2014*; *Perez, Badia & Labarta, 2008*; *Gautier et al., 2013*; *Bauer et al., 2012*; *Tillenius, 2015*; *Augonnet et al., 2011*; *Bramas, 2019b*) and each of them has its own specificity, capabilities and interface. Moreover, the well-known and

Corresponding author
Bérenger Bramas,
berenger.bramas@inria.fr

widely used OpenMP standard (*OpenMP Architecture Review Board, 2013*) also supports the tasks and dependencies paradigm since version 4. The advantage of the method to achieve high-performance and facilitate the use of heterogeneous computing nodes has been demonstrated by the development of many applications in various fields (*Sukkari et al., 2018*; *Moustafa et al., 2018*; *Carpaye, Roman & Brenner, 2018*; *Agullo et al., 2016*; *Agullo et al., 2017*; *Agullo et al., 2015*; *Myllykoski & Mikkelsen, 2019*).

However, multiple challenges remain open to bring the task-based approach to non-HPC experts and to support performance portability. In our opinion, the two main problems on a single computing node concern the scheduling and granularity. The scheduling is the distribution of the tasks over the processing units, i.e., the selection of a task among the ready ones and the choice of a processing unit. This is a difficult problem, especially when using heterogeneous computing nodes as it cannot be solved optimally in general. Much research is continuously conducted by the HPC and the scheduling communities to provide better generic schedulers (*Bramas, 2019a*). The granularity issue is related to the size of the tasks. When the granularity is too small, the overhead of task management, and the potential data movements, becomes dominant and can dramatically increase the execution time due to the use of synchronizations (*Tagliavini, Cesarini & Marongiu, 2018*). On the other hand, when the granularity is too large, it reduces the degree of parallelism and leaves some processing units idle. Managing the granularity can be done at different levels. In some cases, it is possible to let the developer adapt the original algorithms, computational kernels, and data structures, but this could require significant programming effort and expertise (*Bramas, 2016*). An alternative, as we aim to study in this paper, is to investigate how to cluster the nodes of task graphs to increase the granularity of the tasks and thus obtain faster execution by mitigating the overhead from the management of the dependencies. An important asset of this approach is that working at the graph level allows creating a generic method independent from the application and what is done at the user level, but also independent of the task-based runtime system that will be used underneath.

While graph partitioning/clustering is a well-studied problem, it is important to note that the obtained meta-DAG (direct acyclic graph) must remain acyclic, i.e., the dependencies between the cluster of nodes should ensure to be executable as a graph of tasks, and keep a large degree of parallelism. Hence, the usual graph partitioning methods do not work because they do not take into account the direction of the edges (*Hendrickson & Kolda, 2000*). Moreover, the DAG of tasks we target can be of several million nodes, and we need an algorithm capable to process them.

In the current study, we use a generic algorithm that has been proposed to solve this problem (*Rossignon et al., 2013*), and we refer to it as the general DAG clustering algorithm (GDCA).

The contributions of the paper are the following:

- We provide two variants of the GDCA, which change how the nodes are aggregated and allow to have clusters smaller than the targeted size;

- We provide a new model to simulate the execution of a DAG, by considering that there are overheads in the execution of each task, but also while releasing or picking a task, and we use this model to find the best clustering size;
- We evaluate and compare DGCA and our approach on a large graph set using emulated executions;
- We evaluate and compare DGCA and our approach on Chukrut (Conservative Hyperbolic Upwind Kinetic Resolution of Unstructured Tokamaks) (*Coulette et al., 2019*) that computes the transport equation on a 2D unstructured mesh.

The paper is organized as follows. In 'Related Work', we summarize the related work and explain why most existing algorithms do not solve the DAG of tasks clustering problem. Then, in 'Problem Statement and Notations', we describe in detail the DAG of tasks clustering problem. We introduce the modifications of the GDCA in 'DAG of Tasks Clustering'. Finally, we evaluate our approach in 'Experimental Study'.

The source code of the presented method and all the material needed to reproduce the results of this paper are publicly available[1].

[1] An implementation of our method is publicly available at https://gitlab.inria.fr/bramas/dagpar. Besides, we provide the test cases used in this paper at https://figshare.com/projects/Dagpar/71198.

## RELATED WORK

Partitioning or clustering usually refers to dividing a graph into subsets so that the sum of costs on edges between nodes in different subsets is minimum. However, our objective here is not related to the costs of the edges, which we consider null, but to the execution time of the resulting graph in parallel considering a given number of threads and runtime overhead. Hence, while it is generally implicit that partitioning/clustering is related to the edge cut, we emphasize that it should be seen as a graph symbolic transformation and that the measure of quality and final objective differ depending on the problem to solve.

Partitioning tends to refer to finding a given number of subgraphs, which is usually much lower than the number of nodes. In fact, once a graph is partitioned, it is usually dispatched over different processes and thus there must be as many subgraphs as there are processes, whereas clustering is more about finding subgraphs of a given approximate size or bounded by a given size limit, where nodes are grouped together if it appears that they have a certain affinity. This is a reason why the term clustering is also used to describe algorithms that cluster indirect graphs by finding hot spots (*Schaeffer, 2007*; *Xu & Tian, 2015*; *Shun et al., 2016*). Moreover, the size of a cluster is expected to be much lower than the number of nodes.

The granularity problem of the DAG of tasks with a focus on the parallel execution has been previously studied. Sarkar and Hennessy designed a method to execute functional programs at a coarse granularity because working at fine granularity, i.e. at the instruction level, was inefficient on general purpose multiprocessors (*Sarkar & Hennessy, 1986*; *Sarkar, 1989*). They proposed a compile-time clustering approach to achieve the trade-off between parallelism and the overhead of exploiting parallelism and worked on graphs obtained directly from the source code. As we do in the current paper, they focused on the performance, i.e. best execution time, as a measure of the quality of the clustering and their estimation of execution times were based on the number of processors, communication

and scheduling overhead. However, their clustering algorithm is different from ours. It starts by considering each node as a cluster and successively merges them until it obtains a single subgraph while keeping track of the best configuration found so far to be used at the end. By doing so, their algorithm has above quadratic complexity and thus is unusable to process very large graphs. Also, in our case, we do not take communication into account, and we consider that some parts of the scheduling overhead are blocking: no threads can peek a task when another thread is already interacting with the scheduler.

More recently *Rossignon et al. (2013)* proposed GDCA to manage DAG of fine grain tasks on multi-core architectures. Their first solution is composed of three main algorithms called *sequential*, *front* and *depth front*. The sequential algorithm puts together a task that has only one predecessor with its parent. The front algorithm reduces the width of the graph at each level. The depth front performs a breadth-first traversal of the descendants of a task to aggregate up to a given number of tasks together. They extended this last algorithm (*Rossignon, 2015*) by proposing a generic method, GDCA, that we use in the current study[2]. The authors also provided a new execution model to simulate the execution of a DAG where they use an overhead per task and a benefit coefficient for aggregation.

In a more general context, the community has focused on indirect graph partitioning. A classical approach, called the two-way partitioning, consists in splitting a graph into two blocks of roughly equal size or in minimizing the edge cut between the two blocks (*Kernighan & Lin, 1970*; *Fiduccia & Mattheyses, 1982*). The method can be applied recursively multiple times until the desired number of subgraphs is reached. Later, multi-way methods have been proposed (*Hendrickson & Leland, 1995*; *Karypis & Kumar, 1998*; *Karypis et al., 1999*) and most of them have been done in the context of very-large-scale integration (VLSI) in an integrated circuit. The motivation is to partition large VLSI networks into smaller blocks of roughly equal size to minimize the interconnections between the blocks. The multi-way partitioning has been improved by taking into account the direction of the edges in the context of Boolean networks (*Cong, Li & Bagrodia, 1994*). The authors showed that considering the direction of the edges is very helpful, if not mandatory, in the design in order to have acyclic partitioning.

The problem of acyclic DAG partitioning has also been studied by solving the edge-cut problem, i.e., by minimizing the number of weights of the edges having endpoints in different parts and not by focusing on the execution time (*Herrmann et al., 2017*). We argue that the execution time is the only criteria that should be evaluated and that measuring the edge-cut coefficient is not accurate to estimate the benefit of the clustering.

Other studies have focused on partitioning with special constraints, such as finding a minimum cost partition of the graph into subsets of size less than or equal to a criteria (*Kernighan, 1971*), which can be seen as dividing a program into pages of fixed size to minimize the frequency of inter-page transitions. The problem also exists with FPGA, where a complete program does not fit in the field and thus should be divided in sub-parts with the objective of minimizing the re-programming (*Purna & Bhatia, 1999*). In linear algebra, the LU factorization can be represented as a tree graph that can be partitioned in linear time (*Pothen & Alvarado, 1992*).

[2] The thesis that includes this final version is written in French.

The real application we used to assess our method solves the transport equation on unstructured meshes. Task-based implementations to solve the transport equation on a grid (i.e. structured and regular mesh) have already been proposed by *Moustafa et al. (2018)*. The authors have created a version on top of the ParSEC runtime system where they partitioned the mesh and avoided working on the graph of tasks. Working on the mesh is another way to partition the graph, but this was possible in their case because the dependencies on the mesh were known and regular. The dependencies were not impacted by the clustering because an inter-partition dependency would simply be transformed into the exchange of a message. In other words, a process could work on its sub-graph even if some of its nodes are pending for input that will be sent by other processes. This is quite different from our approach, as we consider that a sub-graph is transformed into a macro-task, and hence all input dependencies must be satisfied before a thread starts to work on it.

## PROBLEM STATEMENT AND NOTATIONS

Consider a DAG $G(V,E)$ where the vertices $V$ are tasks and the edges $E$ are the dependencies between those. The clustering problem of a DAG of tasks consists in finding the best clusters to obtain the minimal makespan possible when the DAG is executed on a specific hardware or execution model. Implicitly, the hardware or execution model should have some overheads, which could come from the management of the tasks for example, or the minimal execution time will always be obtained without clustering, i.e. on irrealistic hardware without overhead, any clustering of tasks will reduce the degree of parallelism without offering any advantages. Finding the optimal solution is NP-complete (*Johnson & Garey, 1979*) because it requires to test all the possible combinations of clusters. Moreover, evaluating a solution is performed by emulating a parallel execution, which has a complexity of $\mathcal{O}(V \log(W) + E)$, where $W$ is the number of workers and usually considered constant.

In this paper, we solve a sub-problem that we call the clustering problem of a DAG of tasks with no-communications since we consider that the weights of the edges are insignificant and the edges are only here to represent the dependencies. This problem is met when moving data has no cost or is negligible, which is the case if the workers are threads and the NUMA effects negligible or if we use processes but have a way to hide communication with computation.

Classical partitioning algorithms for indirected graphs cannot be used because they will not obtain an acyclic macro-graph. Formally, a macro-graph remains acyclic if for any edge $a \rightarrow b$ the corresponding clusters $\mathcal{C}(a) \leq \mathcal{C}(b)$ (note that $\mathcal{C}(a) = \mathcal{C}(b)$ means that $a$ and $b$ are in the same cluster). This is also know as the convexity constraint (*Sarkar & Hennessy, 1986*) where we say that a subgraph $H$ of graph $G$ is convex if any path $\mathcal{P}(a,b)$, with $a$, $b$ $\in H$, is completely contained in $H$. Consequently, one way to solve the problem would be to find a valid topological order of the nodes and divide it into clusters.

We write $M$ the desired size of clusters, which should be seen as an upper limit such that no cluster will have more than $M$ nodes.

**Parallel efficiency and edge cut**. There is a relation between the edge-cut and the parallel execution when the edges represent communication costs between cluster owners.

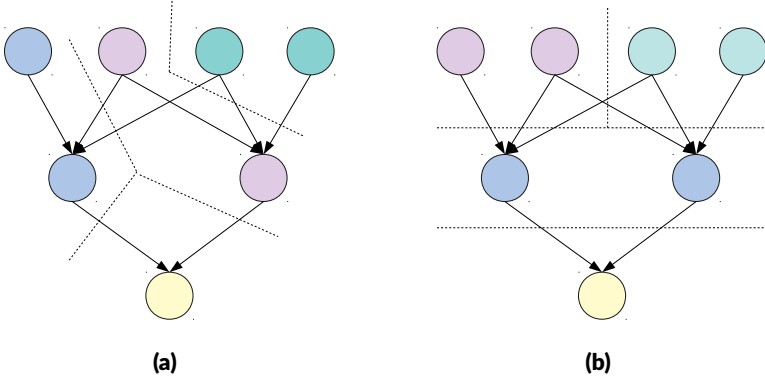

**Figure 1 Example of clustering a DAG of 7 nodes targeting cluster of $M = 2$ nodes.** If each edge has a weight of 1, the cut cost is 7 for (A) and 8 for (B). If each node is a task of cost 1 and edges are not taken into account, the parallel execution duration is 7 units for (A) and 5 units for (B). If each node is a task of cost 1 and edges are considered as communication of 1 unit sent sequentially after completion of the cluster, the parallel execution duration is 11 units for (A) and 9 units for (B).

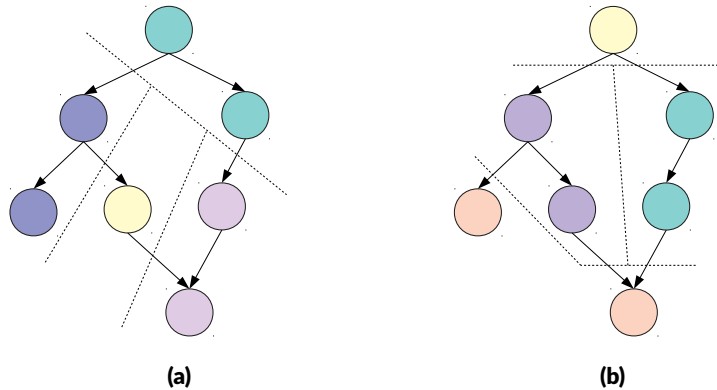

**Figure 2 Example of clustering a DAG of 7 nodes targeting cluster of $M = 2$ nodes.** If each edge has a weight of 1, the cut cost is 4 for (A) and 5 for (B). If each node is a task of cost 1 and edges are not taken into account, the parallel execution duration is 7 units for (A) and 5 units for (B). If each node is a task of cost 1 and edges are considered as communication of 1 unit sent sequentially after completion of the cluster, the parallel execution duration is 10 units for (A) and 6 units for (B).

However, looking only at the edge-cut is not relevant when the final and only objective is the parallel efficiency. Moreover, this is even truer in our case because the weights of the edges are neglected. To illustrate the differences, we provide in Figs. 1 and 2 examples that demonstrate that when attempting to obtain a faster execution, the edge-cut is not the most important feature. In both examples, the configuration with the lowest edge-cut is slower when executed in parallel whether communications are taken into account or not. **Clustering and delay in releasing dependencies.** Traditionally, graph partitioning is used to distribute the workload on different processes while trying to minimize communications. In our case, however, a cluster is a macro-task and is managed like a task: it has to wait

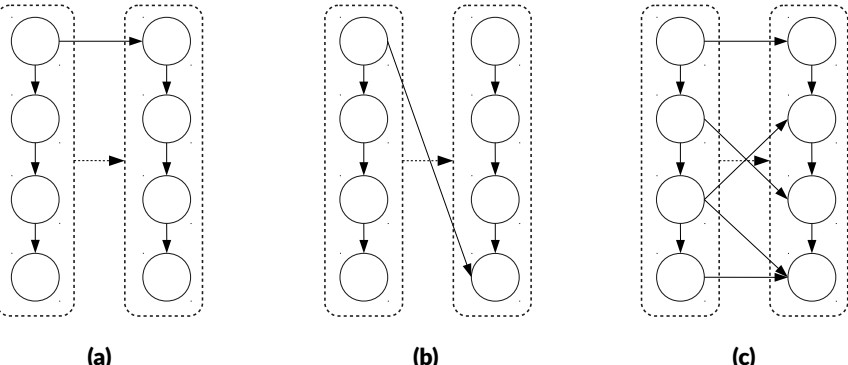

**Figure 3** **Example of clustering three DAGs of eight nodes targeting in two cluster of $M = 4$ nodes.** The obtained meta-DAG is the same despite the original dependencies between the nodes, and the cluster on the right will have to wait that the cluster on the left is fully executed to become ready. (A) Graph with a dependency between the two first nodes. (B) Graph with dependencies between first and last nodes. (C) Graph with multiple dependencies.

for all its dependencies to be released to become ready, and it releases its dependencies once it is entirely completed and not after the completion of each task that composes it. This means the number of dependencies that exists originally between the tasks of two macro-tasks is not relevant, because if there is one or many then the two macro-tasks are linked, as illustrated by the Fig. 3. A side effect is that creating macro-tasks delays the release of the dependencies. In fact, the release of the dependencies can be delayed by the complete duration of the macro-task compared to execution without clustering and this delay also implies a reduction of the degree of parallelism. However, if the degree of parallelism at a global level remains high, the execution could still be faster because it is expected that the clustering will reduce the overhead.

## DAG OF TASKS CLUSTERING

### Modification of the GDCA

Before entering into the details of our approach, we first give a general description of the original algorithm. The algorithm continuously maintains a list of ready tasks by processing the graph while respecting the dependencies. It works on the ready tasks only to build a cluster. By doing so, all the predecessors of the ready tasks are already assigned to a cluster, and all the ready tasks and their successors are not assigned to any cluster. This strategy ensures that no cycle will be created while building the clusters. To create a cluster, the algorithm first picks one of the ready tasks, based on a heuristic that we call the initial-selection. Then it iteratively aggregates some ready nodes to it until the cluster reaches $M$ nodes, using what we call aggregate-selection. Every time a node is put in a cluster, the algorithm releases its dependencies and potentially adds new ready tasks to the list.

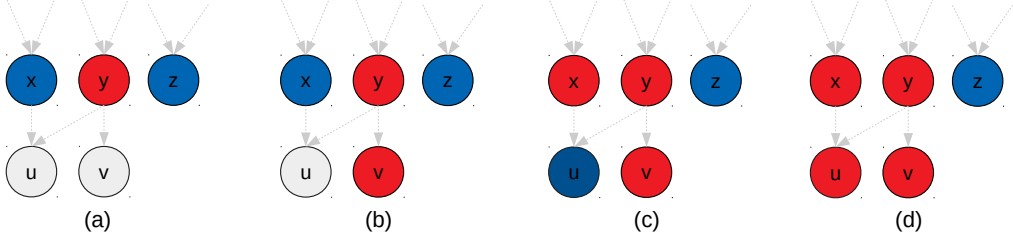

**Figure 4  Illustration of the GDCA.** Ready tasks are in blue, tasks assigned to the new cluster are in red. (A) Nodes $x$, $y$ and $z$ are ready, and here $y$ is selected as first node for the current cluster $p$ to create (initial-selection). (B) Nodes $x$, $v$ and $z$ are ready, and we have to select the next nodes to put in $p$ (agregate-selection). If the criteria to decide is the number of predecessors in the cluster, then $v$ is selected. (C) Nodes $x$ and $z$ are ready, and both nodes have zero predecessors in $p$. If we look at the successors they have in common with $p$, then $u$ is selected. (D) node $z$ is ready, and it might have no predecessors or successors in common with nodes in $p$. If we use a strict cluster size, then $z$ should be added to $p$, otherwise, the cluster $p$ is done.

The initial-selection and the aggregate-selection are the two heuristics that decide which node to select to start a cluster, and which nodes to add to the ongoing cluster. The initial-selection picks the node with the lowest depth, where the depth of a node is the longest distance to the roots. The aggregate-selection picks the tasks that has the largest number of predecessors in the new cluster. For both selections, the lowest ids is selected in case of equality to enforce a strict order of the nodes.

In Figure 4, we provide an example of clustering that illustrates why we potentially need additional heuristics for the aggregate-selections. In Fig. 4B both nodes $x$ and $z$ are ready and could be selected, but node $z$ has no connections with the new cluster. The original algorithm does not have a mechanism to detect this situation. Second, in Fig. 4D since $z$ has no connections with the new cluster, it could be disadvantageous to add it. If we imagine the case where a graph is composed of two independent parts, connecting them is like putting a synchronization on their progress. On the other hand, if we need clusters of a fixed size, as it is the case in the original algorithm, there is no choice and $z$ must be put in the cluster.

In Appendix, we provide two graphs that were clustered using GDCA, see Figs. A1 and A2.

**Change in the initial-selection.** We propose to select the node using the depth (like in the original algorithm), but also to use the number of predecessors. Our objective is to privilege the nodes with the highest number of predecessors that we consider more critic.

**Change in the aggregate-selection.** To select the next node to add to a cluster, we choose the node with the highest number of predecessors in the cluster, but in case of equality, we compare the number of nodes in common between the successors of the cluster and the successors of the candidates. For instance, in Fig. 4A, the node $x$ has one common successor with the new cluster (node $u$), but as the number of predecessors in the cluster is more significant $v$ is selected. Then, in Fig. 4B, $x$ has one common successor, and $z$ none, therefore, with this heuristic $x$ is selected.

**Flexible cluster size.** If no nodes in the ready list have a predecessor in the cluster or a common successor, then we can decide to stop the construction of the current cluster and start a new one, which means that some clusters will have less than $M$ nodes. This heuristic would stop the construction of the new cluster in Fig. 4D.

**Full algorithm.** The complete solution is provided in Algorithm 1. The GDCA algorithm is annotated with our modifications and we dissociate GDCAv2 that includes the change in the nodes selections, and GDCAws that includes the stop when no ready node has a connection with the new cluster.

The main loop, line 6, ensures that the algorithm continues until there are no more ready nodes in the list. The initial-selection is done at line 12, where the algorithm compares the depth, the number of predecessors and the ids of each node. This can be implemented using a sort or simply by selecting the best candidate as the list will be rearranged and updated later in the algorithm. At line 16, the dependencies of the master node are released. If some of its predecessors become ready (line 18), they are put in the list and their counters of predecessors in the new cluster are incremented. Otherwise (line 21), they are put in a set that includes all the successors of the new cluster. This set is used line 27, to count the common successors between the cluster and each ready node. In an optimized implementation, this could be done only if needed, i.e. if the best nodes have equal number of predecessors in the cluster during the aggregate-selection. At line 33, the ready nodes are sorted using their number of predecessors in the cluster, their number of common successors with the cluster and their ids. If we have flexible cluster size, we can stop the construction of the new cluster (line 35) if we consider that no nodes are appropriate. Otherwise, the next node is added to the cluster, its dependencies are released and the counter updated (from line 38 to line 38).

In Appendix, we provide an example of emulated execution of a DAG using this method, see Fig. A3.

## Emulating the execution of DAG of tasks

Iterating on a DAG to emulate an execution with a limited number of processing units is a classical algorithm. However, how the overhead should be defined and included in the model is still evolving (*Kestor, Gioiosa & Chavarra-Miranda, 2015*). We propose to take into account three different overheads: one overhead per task execution, and two overheads for the release and the selection of a ready task.

Using an overhead per task is classically admitted in the majority of models. In our case, this overhead is a constant per task - no matter the task's size - and only impacts the worker that will execute the task. For example, if the overhead per task is $O_t$ and a worker starts the execution of a task of duration $d$ at time $t$, the worker will become available at $t + d + O_t$. The accumulated cost duration implied by this overhead decreases proportionally with the number of tasks (i.e., the more tasks per cluster the less total overhead).

Second, our model includes an overhead every time a task is released or assigned to a worker. When a task is released, it has to be pushed in the ready task list, and this implies either some synchronization or lock-free mechanisms with a non-negligible cost. The same happens when a task is popped from the list and assigned to a worker. Moreover, the

pushes and pops compete to modify the ready list, and in both cases, this can be seen as a lock with only one worker at a time that accesses it. As a result, in our model, we increment the global timestamp variable every time the list is modified.

We provide our solution in Algorithm 2, which is an extension of the DAG execution algorithm with a priority queue of workers. The workers are stored in the queue based on their next availability and we traverse the graph while maintaining the dependencies and a list of ready tasks. At line 5, we initialize the current time variable using the number of ready tasks and the push overhead, considering that all these tasks are pushed in the list and this makes it impossible to access it. Then, at line 8, we assign tasks to the available workers and store them in the priority queues using the cost of each task and the overhead per task, see line 12. Then, in the core loop, line 16, we pick the next available worker, release the dependencies for the task it was computed, and assign tasks to idle workers (until there are no more tasks or idle workers). Finally, we wait for the last workers to finish at line 34.

Our execution model is used to emulate the execution of a DAG on a given architecture, but we also use it to found the best cluster granularity. To do so, we emulate executions starting with a size $M = 2$ and we increase $M$ and keep track of the best granularity found so far $B$. We stop after granularity $2 \times B$. The idea is that we will have a speedup as we increase the granularity until we constraint too much the degree of parallelism. But in order not to stop at the first local minima (the first time an increase of the granularity results in an increase of the execution time), we continue to test until the granularity equals two times the best granularity we have found.

## EXPERIMENTAL STUDY

We evaluate our approach on emulated executions and on a real numerical application.

### Emulated executions

**Graph data-set.** We use graphs of different properties that we summarize in Table 1. They were generated from the Polybench collection (*Grauer-Gray et al., 2012*), the daggen tool (*Suter, 2013*) or by ourselves. The graphs from daggen are complex in the sense that their nodes have important number of predecessors/successors and that the cost of the tasks are significant and of large intervals. The graphs with names starting by *Chukrut* are the test cases for the real application.

**ALGORITHM 1:** GDCA algorithm, where $M$ is the desired cluster size. GDCAv2 includes the lines in **black underlined-bold**. GDCA-ws includes the lines in gray.

```
1  function cluster(G = (V, E), M)
2      ready ← Get_roots(G) // Gets the roots
3      depths ← Distance_from_roots(G, ready) // Gets the distance from roots
4      count_deps_release = ∅ // # of released dependencies
5      cpt_cluster = 0
6      while ready is not empty do
7          count_pred_master = ∅ // # predecessors in the new cluster
8          count_next_common = ∅ // # common successors
9          // Sort by, first increasing depths, second decreasing number
10         // of predecessors, third increasing ids (to ensure a strict
11         // order)
12         ready.sort()
13         master = ready.pop_front()
14         clusters[master] = cpt_cluster
15         master_boundary = ∅
16         for u ∈ successors[master] do
17             count_deps_release[u] += 1
18             if count_deps_release[u] equal |predecessors[u]| then
19                 ready.insert(u)
20                 count_pred_master[u] = 1
21             else
22                 master_boundary.insert(u)
23             end
24         end
25         cluster_size = 1
26         while cluster_size < M do
27             for u ∈ ready do
28                 count_next_common[u] = | successors[u] ∩ master_boundary |
29             end
30             // Sort by, first decreasing count_pred_master, second
31             // increasing depths, third decreasing count_next_common,
32             // fourth increasing ids (to ensure a strict order)
33             ready.sort()
34             next = ready.front();
35             if count_pred_master[next] is 0 AND count_next_common[next] is 0 then
36                 break;
37             end
38             ready.pop_front()
39             cluster_size += 1
40             clusters[next] = cpt_cluster
41             for u ∈ successors[next] do
42                 count_deps_release[u] += 1
43                 if count_deps_release[u] equal |predecessors[u]| then
44                     ready.insert(u)
45                     count_pred_master[u] = 0
46                     for v ∈ predecessors[u] do
47                         if clusters[v] == clusters[master] then
48                             count_pred_master[u] += 1
49                         end
50                     end
51                     master_boundary.erase(u)
52                 else
53                     master_boundary.insert(u)
54                 end
55             end
56         end
57         cpt_cluster += 1
58     end
```

**Hardware.** We consider four systems in total, two imaginary hardware with two type of overhead low (L) and high (H), with the following properties:

---

**ALGORITHM 2:** Emulate an execution of $G$ using $W$ workers. The overheads are *push_overhead*, *pop_overhead* and *task_overhead*.

```
 1  function Emulate_execution(G = (V, E), W, push_overhead, pop_overhead, task_overhead)
 2      idle_worker ← list(0, W-1)
 3      current_time ← 0
 4      ready ← Get_roots(G)
 5      current_time ← push_overhead × ready.size()
 6      nb_computed_task ← 0
 7      workers ← empty_priority_queue()
 8      while ready is not empty AND idle_worker is not empty do
 9          task ← ready.pop()
10          worker ← idle_worker.pop()
11          current_time ← current_time + pop_overhead
12          workers.enqueue(worker, task, current_time, costs[u] + task_overhead)
13          nb_computed_task ← nb_computed_task + 1
14      end
15      deps ← 0
16      while nb_computed_task ≠ |tasks| do
17          [task, worker, end] ← workers.dequeue()
18          current_time ← max(current_time, end)
19          idle_worker.push(worker)
20          for v ∈ successors[u] do
21              deps[v] ← deps[v] + 1
22              if |deps[v]| = |predecessors[v]| then
23                  ready.push(v)
24              end
25          end
26          while ready is not empty AND idle_worker is not empty do
27              task ← ready.pop()
28              worker ← idle_worker.pop()
29              current_time ← current_time + pop_overhead
30              workers.enqueue(worker, task, current_time, costs[u] + task_overhead)
31              nb_computed_task ← nb_computed_task + 1
32          end
33      end
34      while nb_computed_task ≠ |tasks| do
35          [task, worker, end] ← workers.dequeue()
36          current_time ← max(current_time, end)
37      end
38      return current_time
```

---

- Config-40-L: System with 40 threads and overhead per task 0.1, per push 0.2 and per pop 0.2.
- Config-40-H: System with 40 threads and overhead per task 2, per push 1 and per pop 1.
- Config-512-L: System with 512 threads and overhead per task 0.1, per push 0.2 and per pop 0.2.
- Config-512-H: System with 512 threads and overhead per task 4, per push 2 and per pop 2.

The given overheads are expressed in terms of proportion of the total execution time of a graph. Consequently, if $D$ is the total duration of a graph (the sum of the duration of the $N$ tasks), and if the overhead is $O_t$, then the overhead per task is given by $D \times O_t/N$.

**Increasing granularity.** In Figure 5, we show the duration of the emulated executions as the granularity increases for twelve of the graphs. As expected the execution time decreases as the granularity increases in most cases since the impact of the overhead is mitigated. Using 512 threads (Config-512-L/red line) instead of 40 (Config-40-L/green line) does not speed up the execution when the overhead is low, and this is explained by the fact that the average degree of parallelism is lower than 512 for most graphs. In Figs. 5H and 5K, the

execution time shows a significant variation depending on the clustering size. This means that for some cluster sizes several workers are not efficiently used by being idle (waiting for another a worker to finish its task and release the dependencies), while for some other sizes the workload per worker is well balanced and the execution more efficient. Note that we increase the granularity up to two times the best granularity we have found, and this appears to be a good heuristic to catch the best cluster size without stopping at the first local minima.

**Details.** In Table 2, we provide the best speedup we have obtained by taking execution times without clustering as reference. We provide the results for the GDCA and the updated method (GDCAv2), and also show the best granularity for each case. The GDCAv2 provides a speedup over GDCA in many cases. For instance, for the daggen's graphs the GDCAv2 method is faster in most cases. In addition, there are cases where the GDCAv2 provide a significant speedup, such as the graphs *polybench - kernel trmm*, *polybench - kernel jacobi 2d imper* and *polybench - kernel gemvr*. For the Config-512-H configuration and the *polybench - kernel gemvr* graph, the GDCA has a speedup of 24.73, and GDCAv2 83.47. However, there are still many cases where GDCA is faster, which means that to cluster a graph from a specific application it is required to try and compare both methods. In addition, this demonstrates that while our modifications of GDCA seem natural when we look at the graph at a low level, they do not necessarily provide an improvement at a global level due to corner cases. Moreover, the ids of the nodes are more important in GDCA than in GDCAv2, and this information is usually not random and includes the order of construction of the tasks.

Concerning GDCAws, the method is not competitive for most of the graphs. However, it provides a significant speedup for the *Chukrut* graphs, which are the ones use in our real application.

## Real application

### Hardware configuration

We use the following computing nodes:

- Intel-24t : 4 × Intel(R) Xeon(R) CPU E5-2680 v3 @ 2.50GHz, with caches L1 32K, L2256K and L3 15360K (24 threads in total).
- Intel-32t : 2 × Intel(R) Xeon(R) Gold 6142 CPU @ 2.60GHz V4, with caches L1 32K, L21024K and L3 22528K (32 threads in total).

**Software configuration.** We use the GNU C compiler 7.3.0. We parallelize using OpenMP threads and we do not use any mutex during the execution. We only use lock-free mechanisms implemented with C11 atomic variables to manage the dependencies and the list of ready tasks, which is actually an array updated using atomics.

**Test cases.** The two test cases represent a 2D mesh that has the shape of a disk with sizes 40 and 60. The details of the two corresponding graphs are provided in Table 1 under the names *Chukrut - disque40* and *Chukrut - disque60*. The execution of a single task takes around $1.5 \cdot 10^{-5} s$. To estimate the overhead, we take the execution time in sequential $T_1$ and the execution time using a third of the available threads $T_x$ and do

Bramas and Ketterlin (2020), *PeerJ Comput. Sci.*, DOI 10.7717/peerj-cs.247

**Table 1** **Details of the studied graphs.** The degree of parallelism is obtained by iterating on the graph, while respecting the dependencies, and measure the size of the ready task list (the average size or the largest size found during the execution).

| Name | #vertices | #edges | #Predecessors | | Total cost | Cost | | | Degree of parallelism | |
|---|---|---|---|---|---|---|---|---|---|---|
| | | | avg | max | | min | avg | max | avg | max |
| createdag - agraph-2dgrid-200 | 39999 | 79202 | 1.980 | 2 | 39999 | 1 | 1 | 1 | 100.5 | 398 |
| createdag - agraph-deptree-200 | 10100 | 19900 | 1.970 | 2 | 10100 | 1 | 1 | 1 | 50.5 | 100 |
| createdag - agraph-doubletree-200 | 10100 | 19900 | 1.970 | 2 | 10100 | 1 | 1 | 1 | 50.5 | 100 |
| createdag - agraph-tree-200 | 20100 | 39800 | 1.980 | 2 | 20100 | 1 | 1 | 1 | 100.5 | 200 |
| Chukrut - disque40 | 19200 | 38160 | 1.99 | 2 | 19200 | 1 | 1 | 1 | 80.3347 | 120 |
| Chukrut - disque60 | 43200 | 86040 | 1.99 | 2 | 43200 | 1 | 1 | 1 | 120.334 | 180 |
| daggen - 1000-0 5-0 2-4-0 8 | 1000 | 6819 | 6.819 | 42 | 2.2e+14 | 3.1e+08 | 2.2e+11 | 1.3e+12 | 27.7 | 52 |
| daggen - 128000-0 5-0 2-2-0 8 | 128000 | 11374241 | 88.861 | 549 | 3.0e+16 | 2.6e+08 | 2.3e+11 | 1.4e+12 | 362.6 | 641 |
| daggen - 16000-0 5-0 8-4-0 8 | 16000 | 508092 | 31.756 | 97 | 3.7e+15 | 2.7e+08 | 2.3e+11 | 1.4e+12 | 124.03 | 155 |
| daggen - 4000-0 2-0 8-4-0 8 | 4000 | 6540 | 1.635 | 7 | 9.0e+14 | 2.6e+08 | 2.2e+11 | 1.4e+12 | 7.8 | 15 |
| daggen - 64000-0 2-0 2-4-0 8 | 64000 | 169418 | 2.647 | 31 | 1.5e+16 | 2.6e+08 | 2.3e+11 | 1.4e+12 | 11.84 | 23 |
| polybench - kernel 2 mm | 14600 | 22000 | 1.507 | 40 | 14600 | 1 | 1 | 1 | 286.275 | 600 |
| polybench - kernel 3 mm | 55400 | 70000 | 1.264 | 40 | 55400 | 1 | 1 | 1 | 780.282 | 1400 |
| polybench - kernel adi | 97440 | 553177 | 5.677 | 7 | 97440 | 1 | 1 | 1 | 43.4806 | 86 |
| polybench - kernel atax | 97040 | 144900 | 1.493 | 230 | 97040 | 1 | 1 | 1 | 220.045 | 440 |
| polybench - kernel covariance | 98850 | 276025 | 2.792 | 70 | 98850 | 1 | 1 | 1 | 686.458 | 3500 |
| polybench - kernel doitgen | 36000 | 62700 | 1.742 | 2 | 36000 | 1 | 1 | 1 | 3000 | 3000 |
| polybench - kernel durbin | 94372 | 280870 | 2.976 | 250 | 94372 | 1 | 1 | 1 | 2.96088 | 250 |
| polybench - kernel fdtd 2d | 70020 | 220535 | 3.150 | 4 | 70020 | 1 | 1 | 1 | 1167 | 1200 |
| polybench - kernel gemm | 340200 | 336000 | 0.988 | 1 | 340200 | 1 | 1 | 1 | 4200 | 4200 |
| polybench - kernel gemver | 43320 | 71880 | 1.659 | 120 | 43320 | 1 | 1 | 1 | 179.008 | 14400 |
| polybench - kernel gesummv | 125750 | 125500 | 0.998 | 1 | 125750 | 1 | 1 | 1 | 499.008 | 500 |
| polybench - kernel jacobi 1d imper | 79600 | 237208 | 2.980 | 3 | 79600 | 1 | 1 | 1 | 398 | 398 |
| polybench - kernel jacobi 2d imper | 31360 | 148512 | 4.736 | 5 | 31360 | 1 | 1 | 1 | 784 | 784 |
| polybench - kernel lu | 170640 | 496120 | 2.907 | 79 | 170640 | 1 | 1 | 1 | 1080 | 6241 |
| polybench - kernel ludcmp | 186920 | 537521 | 2.876 | 80 | 186920 | 1 | 1 | 1 | 53.7126 | 6480 |
| polybench - kernel mvt | 80000 | 79600 | 0.995 | 1 | 80000 | 1 | 1 | 1 | 400 | 400 |
| polybench - kernel seidel 2d | 12960 | 94010 | 7.254 | 8 | 12960 | 1 | 1 | 1 | 62.3077 | 81 |
| polybench - kernel symm | 96000 | 93600 | 0.975 | 1 | 96000 | 1 | 1 | 1 | 2400 | 2400 |
| polybench - kernel syr2k | 148230 | 146400 | 0.988 | 1 | 148230 | 1 | 1 | 1 | 1830 | 1830 |
| polybench - kernel syrk | 148230 | 146400 | 0.988 | 1 | 148230 | 1 | 1 | 1 | 1830 | 1830 |
| polybench - kernel trisolv | 80600 | 160000 | 1.985 | 399 | 80600 | 1 | 1 | 1 | 100.75 | 400 |
| polybench - kernel trmm | 144000 | 412460 | 2.864 | 80 | 144000 | 1 | 1 | 1 | 894.41 | 1410 |

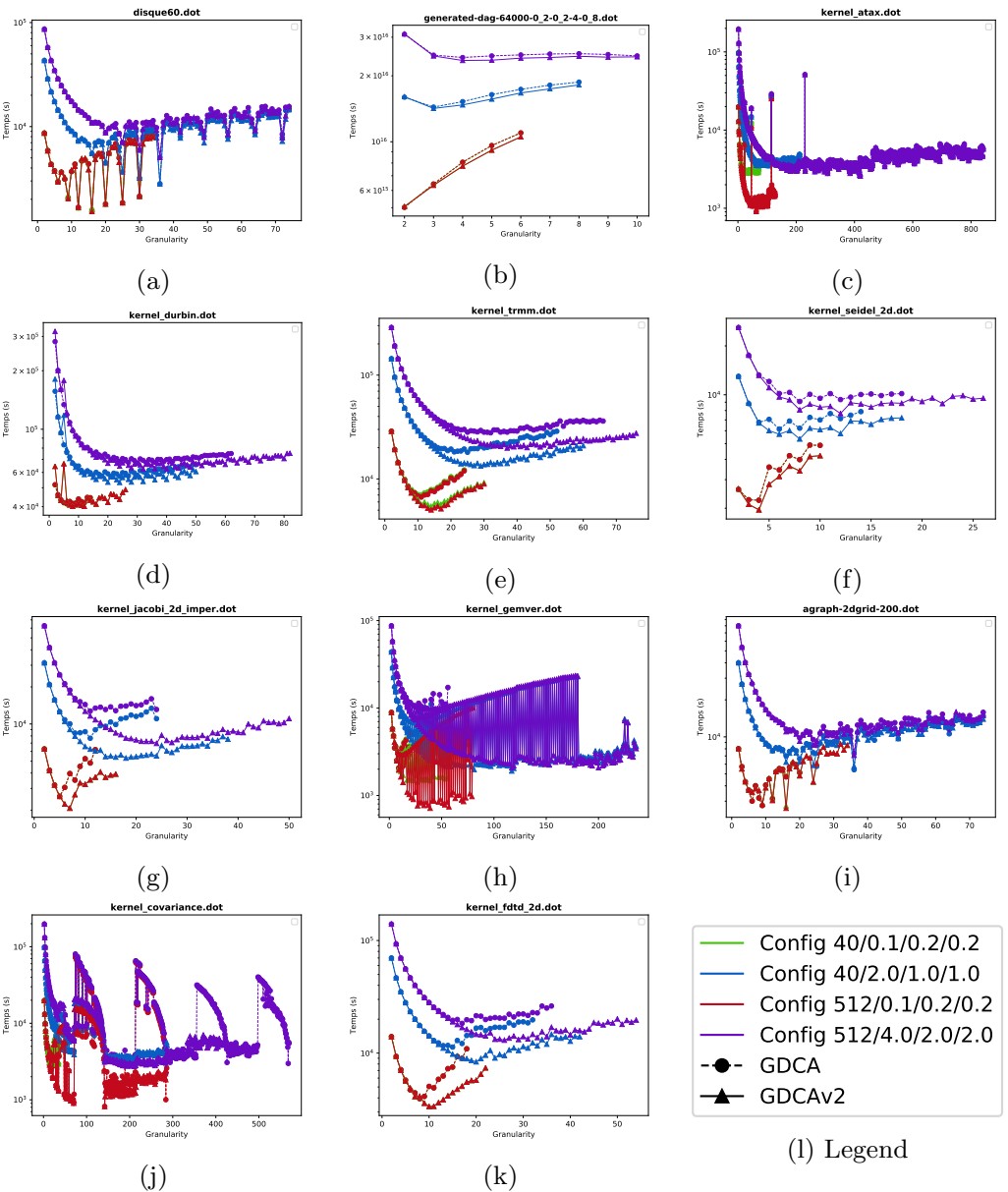

**Figure 5** **Emulated execution times against cluster granularity *G* for different test cases, different machine configurations (colors ----) and different strategies (nodes ● ▲).** (A) disque60. (B) generated-dag-4000-0.2-0.2-4-0.8. (C) kernel atax. (D) kernel durbin. (E) kernel trmm. (F) kernel seidel. (G) kernel jacobi 2D imper. (H) kernel gemver. (i) agraph 2dgrid-200. (J) kernel covariance. (K) kernel fdtd 2D.

$O = (T_x \times x - T_1)/T_1$. We obtained an overhead of 3 on Intel-24t and of 3.5 on Intel-32t. We dispatch this overhead one half for the overhead per task, and one quarter for the push/pop overheads. The execution times are obtained from the average of 6 executions.

We increase the granularity up to $M = 100$ in order to show if our best granularity selection heuristic would miss the best size.

**Results.** In Fig. 6, we provide the results for the two test cases on the two computing nodes.

Bramas and Ketterlin (2020), *PeerJ Comput. Sci.*, DOI 10.7717/peerj-cs.247

**Table 2** **Speedup obtained by clustering the graphs on emulated executions for GDCA and GDCAv2.** The best granularity for the different graphs is provided, and the speedup is in bold when one of the two clustering strategies appears more efficient than the other for a given hardware. GDCAws is slower in most cases, except for the graphs from *createdag* and *Chukrut*. For the configuration Config-512-H, GDCAws get a speedup of 34 for *Chukrut - disque40* and of 44 for *Chukrut - disque60*.

| Name | Config-40-L | | | | Config-40-H | | | | Config-512-L | | | | Config-512-H | | | |
|---|---|---|---|---|---|---|---|---|---|---|---|---|---|---|---|---|
| | GDCA | | GDCAv2 | | GDCA | | GDCAv2 | | GDCA | | GDCAv2 | | GDCA | | GDCAv2 | |
| | *G* | Sp. | *G* | Sp. | *G* | Sp. | *G* | Sp. | *G* | Sp. | *G* | Sp. | *G* | Sp. | *G* | Sp. |
| createdag - agraph-2dgrid-200 | 9 | 5.775 | 16 | **5.821** | 36 | **14.79** | 36 | 13.86 | 9 | 5.786 | 16 | **6.078** | 36 | 21.76 | 36 | **22.57** |
| createdag - agraph-deptree-200 | 6 | **4.158** | 4 | 3.798 | 16 | 12.75 | 16 | 12.75 | 6 | **4.158** | 4 | 3.801 | 25 | **16.6** | 25 | 16.21 |
| createdag - agraph-doubletree-200 | 6 | **4.158** | 4 | 3.798 | 16 | 12.75 | 16 | 12.75 | 6 | **4.158** | 4 | 3.801 | 25 | **16.6** | 25 | 16.21 |
| createdag - agraph-tree-200 | 16 | 7.905 | 16 | 7.905 | 36 | 23.68 | 36 | **24.06** | 16 | 8.978 | 16 | 8.976 | 64 | **37.5** | 64 | 37.47 |
| Chukrut - disque40 | 16 | 7.62 | 16 | 7.62 | 25 | 21.91 | 25 | 21.91 | 16 | 7.629 | 16 | 7.626 | 25 | 23.91 | 25 | 23.91 |
| Chukrut - disque60 | 16 | 10.9 | 16 | 10.9 | 36 | 30.6 | 36 | 30.6 | 16 | 11.35 | 16 | 11.34 | 36 | 34.22 | 36 | 34.22 |
| daggen - 1000-0 5-0 2-4-0 8 | 2 | 1.425 | 2 | **1.575** | 3 | **2.34** | 4 | 2.221 | 2 | 1.425 | 2 | **1.575** | 6 | 2.704 | 5 | **2.858** |
| daggen - 128000-0 5-0 2-2-0 8 | 2 | 1.835 | 2 | **1.872** | 4 | 2.88 | 5 | **2.994** | 2 | 1.839 | 2 | 1.875 | 7 | 3.517 | 6 | **3.689** |
| daggen - 16000-0 5-0 8-4-0 8 | 2 | 1.787 | 2 | **1.806** | 3 | 2.528 | 3 | **2.646** | 2 | 1.786 | 2 | **1.805** | 4 | 2.955 | 4 | **3.1** |
| daggen - 4000-0 2-0 8-4-0 8 | 2 | 1.066 | 2 | **1.08** | 3 | 1.95 | 3 | **1.97** | 2 | 1.066 | 2 | 1.08 | 4000 | 3.998 | 4000 | 3.998 |
| daggen - 64000-0 2-0 2-4-0 8 | 2 | **1.214** | 2 | 1.203 | 3 | 2.085 | 3 | **2.115** | 2 | 1.214 | 2 | 1.203 | 4 | 2.476 | 4 | **2.555** |
| polybench - kernel 2 mm | 18 | 11.71 | 18 | 11.71 | 62 | **41.6** | 57 | 35.88 | 62 | **22.3** | 31 | 20.52 | 124 | **76.65** | 130 | 59.72 |
| polybench - kernel 3 mm | 20 | 13.05 | 20 | 13.05 | 102 | **48.3** | 52 | 43.05 | 51 | **30.82** | 52 | 28.54 | 153 | **97.71** | 154 | 77.7 |

Bramas and Ketterlin (2020), *PeerJ Comput. Sci.*, DOI 10.7717/peerj-cs.247

**Table 2** (*continued*)

| Name | Config-40-L | | | | Config-40-H | | | | Config-512-L | | | | Config-512-H | | | |
|---|---|---|---|---|---|---|---|---|---|---|---|---|---|---|---|---|
| | GDCA | | GDCAv2 | | GDCA | | GDCAv2 | | GDCA | | GDCAv2 | | GDCA | | GDCAv2 | |
| | G | Sp. | G | Sp. | G | Sp. | G | Sp. | G | Sp. | G | Sp. | G | Sp. | G | Sp. |
| polybench - kernel adi | 10 | 6.045 | 10 | **6.161** | 23 | 15.39 | 24 | **16.19** | 10 | 6.045 | 10 | **6.161** | 30 | 20.81 | 34 | **21.43** |
| polybench - kernel atax | 35 | 13.84 | 35 | 13.84 | 105 | 55.41 | 105 | 55.41 | 63 | 42.83 | 63 | 42.83 | 420 | 150.3 | 420 | 150.3 |
| polybench - kernel covariance | 17 | 13.6 | 17 | 13.6 | 68 | 46.93 | 143 | **60.2** | 142 | 48.29 | 142 | 48.29 | 284 | **178.3** | 211 | 144.7 |
| polybench - kernel doitgen | 960 | 14.77 | 960 | 14.77 | 960 | 69.37 | 960 | 69.37 | 120 | 59.98 | 120 | 59.98 | 360 | 188.5 | 360 | 188.5 |
| polybench - kernel durbin | 6 | 1.756 | 12 | **1.806** | 24 | 4.954 | 24 | **5.322** | 6 | 1.752 | 12 | **1.809** | 30 | 7.983 | 40 | **8.594** |
| polybench - kernel fdtd 2d | 8 | 7.116 | 10 | **8.339** | 15 | 12.39 | 20 | **16.74** | 8 | 7.116 | 10 | **8.34** | 17 | 15.26 | 26 | **21.49** |
| polybench - kernel gemver | 9 | 8.349 | 27 | **12.14** | 19 | 17.86 | 117 | **45.44** | 9 | 8.398 | 39 | **24.19** | 27 | 24.73 | 117 | **83.47** |
| polybench - kernel gesummv | 31 | 14.44 | 31 | 14.44 | 114 | 61.21 | 114 | 61.21 | 503 | 83.4 | 503 | 83.4 | 503 | 333.8 | 503 | 333.8 |
| polybench - kernel jacobi 1d imper | 15 | 7.114 | 12 | **8.295** | 32 | 19.84 | 32 | **20.1** | 15 | 7.114 | 12 | **8.295** | 32 | 28.17 | 60 | **28.7** |
| polybench - kernel jacobi 2d imper | 5 | 4.853 | 7 | **6.046** | 11 | 8.184 | 18 | **11.85** | 5 | 4.853 | 7 | **6.046** | 11 | 9.841 | 24 | **18.02** |
| polybench - kernel lu | 16 | **11.47** | 17 | 9.404 | 39 | **24.65** | 39 | 22.85 | 22 | **13.69** | 16 | 11.02 | 86 | 38.65 | 89 | **33.95** |
| polybench - kernel ludcmp | 17 | **6.668** | 13 | 6.379 | 48 | 16.48 | 32 | **16.96** | 17 | **7.366** | 17 | 6.895 | 48 | 24.71 | 60 | **25.38** |
| polybench - kernel mvt | 400 | 15.62 | 400 | 15.62 | 400 | 70.99 | 400 | 70.99 | 200 | 88.87 | 200 | 88.87 | 600 | 280.7 | 600 | 280.7 |
| polybench - kernel seidel 2d | 4 | 2.319 | 4 | **2.659** | 6 | 4.219 | 8 | **4.856** | 4 | 2.319 | 4 | **2.659** | 8 | 5.705 | 12 | **6.772** |
| polybench - kernel symm | 2400 | 15.89 | 2400 | 15.89 | 2400 | 77.36 | 2400 | 77.36 | 200 | 97.94 | 200 | 97.94 | 600 | 308.7 | 600 | 308.7 |
| polybench - kernel syr2k | 27 | 14.24 | 27 | 14.24 | 3726 | 77.85 | 3726 | 77.85 | 324 | 116.9 | 324 | 116.9 | 810 | 383.5 | 810 | 383.5 |
| polybench - kernel syrk | 27 | 14.24 | 27 | 14.24 | 3726 | 77.85 | 3726 | 77.85 | 324 | 116.9 | 324 | 116.9 | 810 | 383.5 | 810 | 383.5 |
| polybench - kernel trisolv | 8 | 4.732 | 8 | 4.732 | 25 | 15.14 | 25 | 15.14 | 9 | 4.87 | 9 | 4.87 | 25 | 18.61 | 25 | 18.61 |
| polybench - kernel trmm | 11 | 8.255 | 14 | **10.6** | 25 | 15.94 | 29 | **21.6** | 11 | 8.622 | 14 | **11.44** | 32 | 20.68 | 37 | **29** |

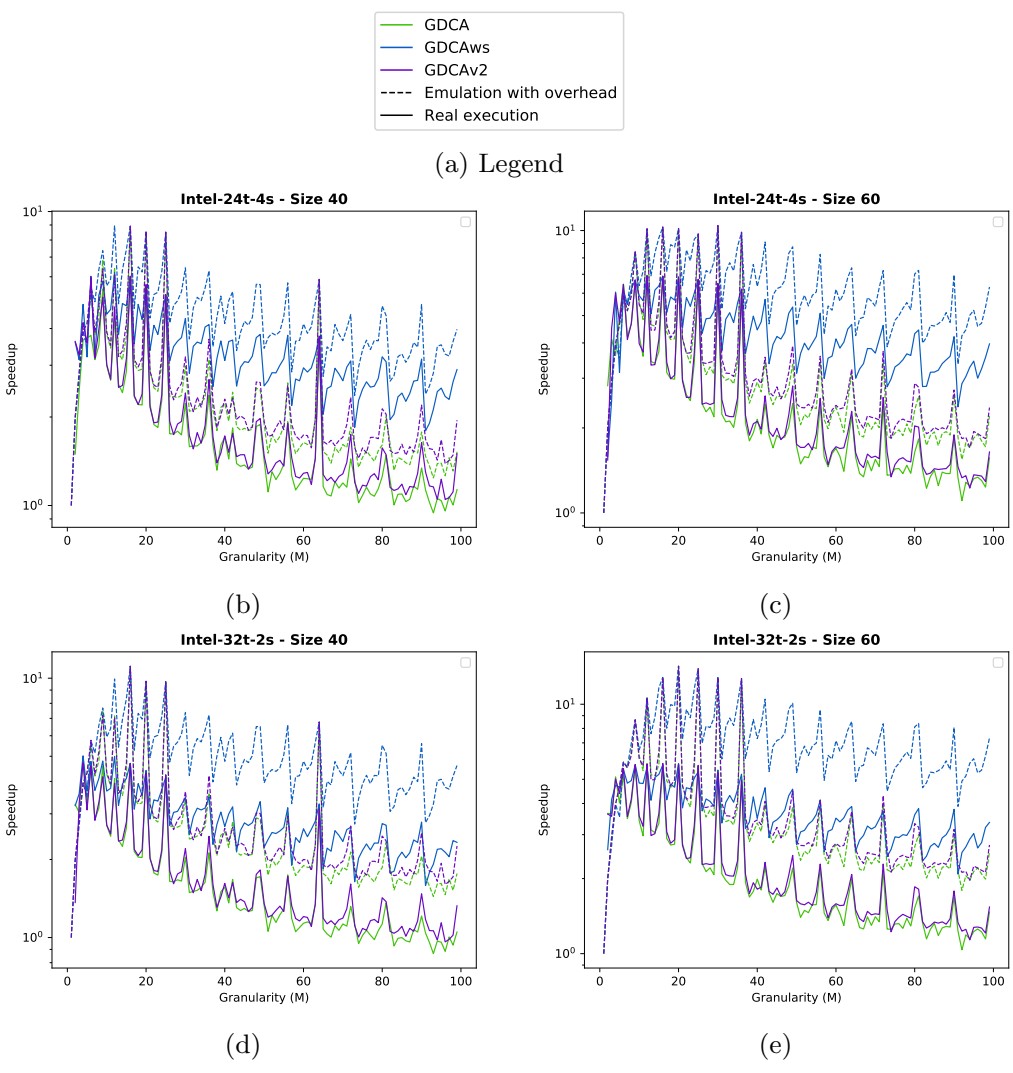

Figure 6  **Speedup obtained for the two test cases with different clustering strategies.** We show the speedup obtained from emulated executions (dashed lines) and from the real executions (plain lines). (A) Inter-24t-4s Size 40. (B) Inter-24t-4s Size 60. (C) Inter-32t-2s Size 40. (D) Inter-32t-2s Size 60.

In the four configurations, the emulation of GDCAws (blue dashed line) is too optimistic compared to the real execution (blue line): on Intel-24t, Figs. 6B and 6C, GDCAws performed poorly (blue line), but on Intel-32t, Figs. 6D and 6E, GDCAws is efficient for large granularities. However, even if it is efficient on average on Intel-32t, it never provides the best execution. This means that having a flexible cluster size is not the best approach for these graphs and that having fewer clusters but of a fixed size (even if it adds more dependencies) seems more appropriate. Concerning the emulation, our model does not catch the impact of having clusters of different sizes.

The emulation of GDCA is more accurate (dashed green line) when we compare it with the real execution (green line), even if the real execution is always underperforming. However, the global shape is correct and, importantly, the performance peaks that happen

in the emulation also happen in the real execution. This means that we can use emulated executions to find the best clustering size for the GDCA. In terms of performance, GDCA provides the best execution on the Intel-32t for $M$ between 10 and 20.

The emulation of GDCAv2 is accurate for the Intel-24t (Figs. 6B and 6C) with a superimposition of the plots (dashed/plain purple lines). However, it is less accurate for the Intel-32t (Figs. 6D and 6E) where the real execution is underperforming compared to the emulation. As for the GDCA, the peaks of the emulation of the GDCAv2 concord with the peaks of the real executions. GDCAv2 provides the best performance on the Intel-24t, for $M$ between 15 and 20.

GDCA and GDCAv2 have the same peaks; therefore, for some cluster sizes the degree of parallelism is much better and the filling of the workers more efficient. But GDCAv2 is always better on the Intel-32t, while on the Intel-24t both are very similar except that GDCA is faster at the peaks. While the difference is not necessarily significant this means that the choice between GDCA and GDCAv2 is machine-dependent.

## CONCLUSION

The management of the granularity is one of the main challenges to achieve high-performance using the tasks and dependencies paradigm. GDCA allows controlling the granularity of the tasks by grouping them to obtain a macro-DAG. In this paper, we have presented GDCAv2 and GDCAws two modified version of GDCA. We evaluated the benefit of GDCA and GDCAv2 on emulated executions. We have demonstrated that our modifications allow obtaining significant speedup in several cases but that it remains unknown which of the two methods will give the best results. We evaluated the three algorithms on a real application and we have demonstrated that clustering the DAG allowed to get a speedup of 7 compared to executions without clustering. We were able to find the best granularity using emulated execution based on a model that incorporates different overheads.

As a perspective, we would like to plug our algorithm directly inside an existing task-based runtime system to cluster the DAG on the fly during the execution. This would require a significant programming effort but will open the study of more applications and certainly lead to improving not only the selection of the parameters but also the estimation of the costs and the overheads. In addition, we would like to adapt the aggregate-selection during the clustering process in order to always use the best of GDCA and GDCAv2.

## ACKNOWLEDGEMENTS

The experiments presented in this paper were carried out using the PlaFRIM experimental testbed, supported by Inria, CNRS (LABRI and IMB), Université de Bordeaux, Bordeaux INP and Conseil Régional d'Aquitaine (see https://www.plafrim.fr/).

## APPENDIX

Figures A1 and A2 are examples of graph clustering with our method. Figure A3 shows an example of a graph clustering and an emulated execution.

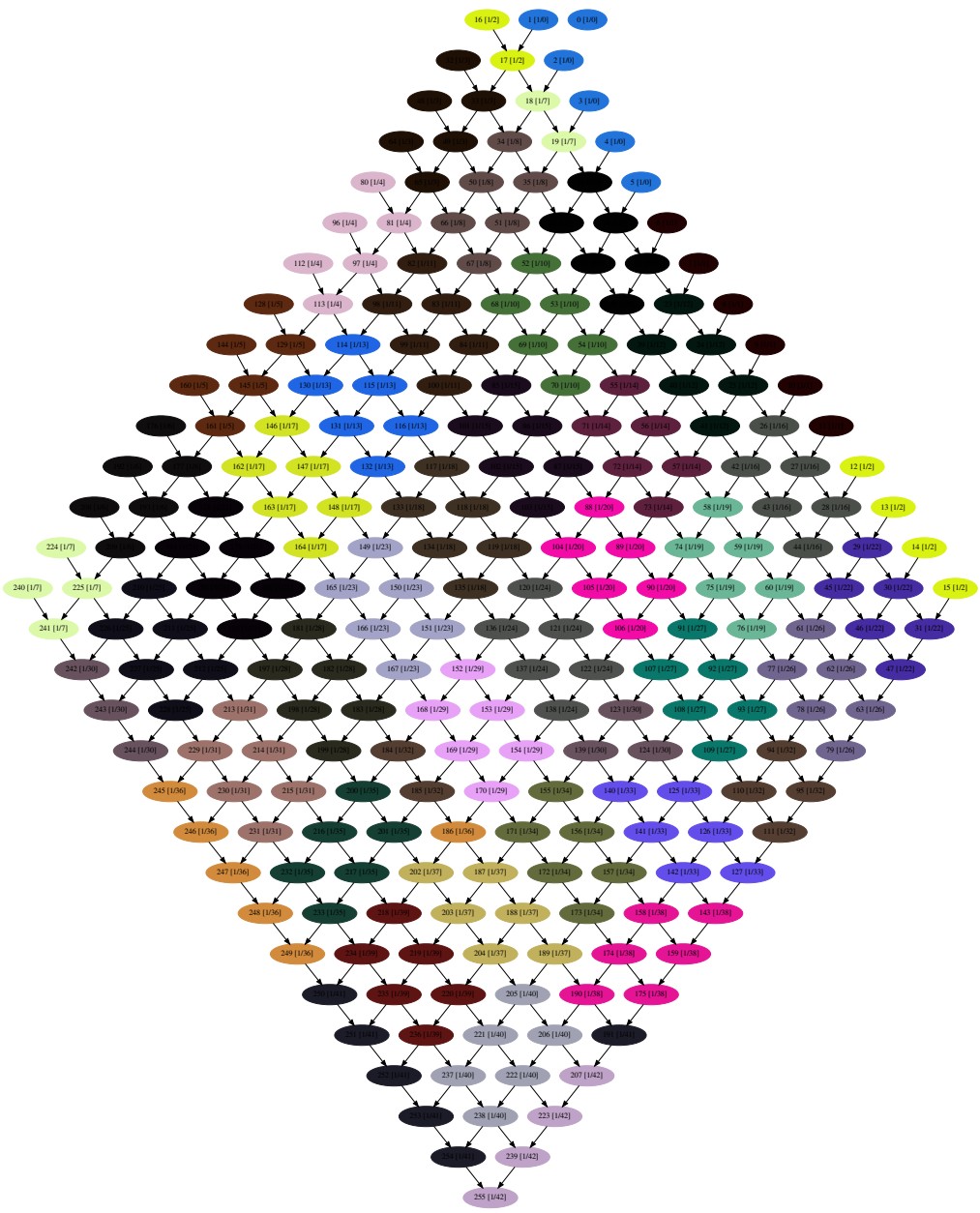

**Figure A1   Clustering of a graph of 256 nodes generated by propagation of the dependencies on a 2D grid from one corner to the opposite one.** The cluster size is $M = 6$. There are 43 clusters.

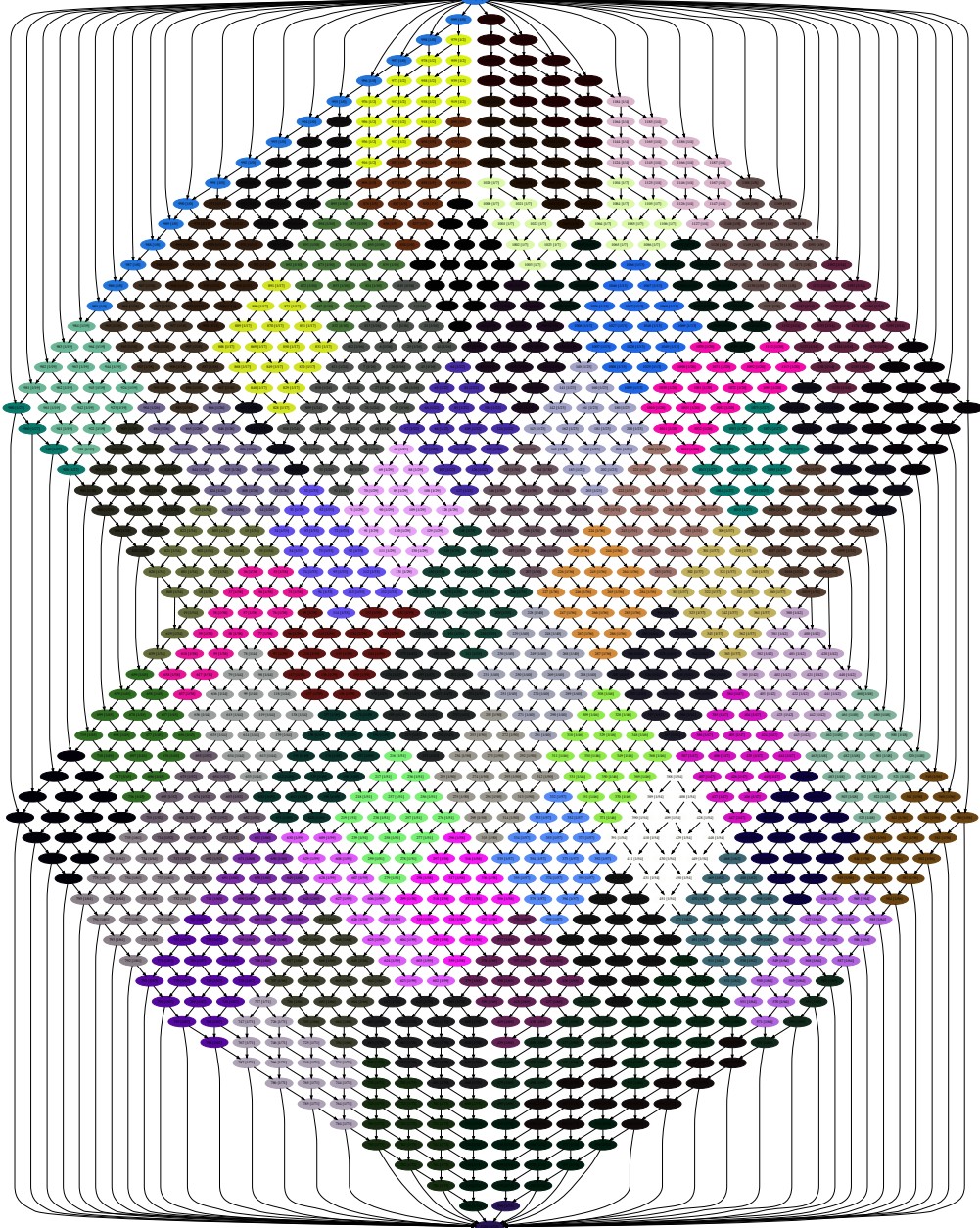

**Figure A2    Clustering of a graph of 1202 nodes generated by the transport equation on a disk.** The cluster size is $M = 16$. There are 76 clusters.

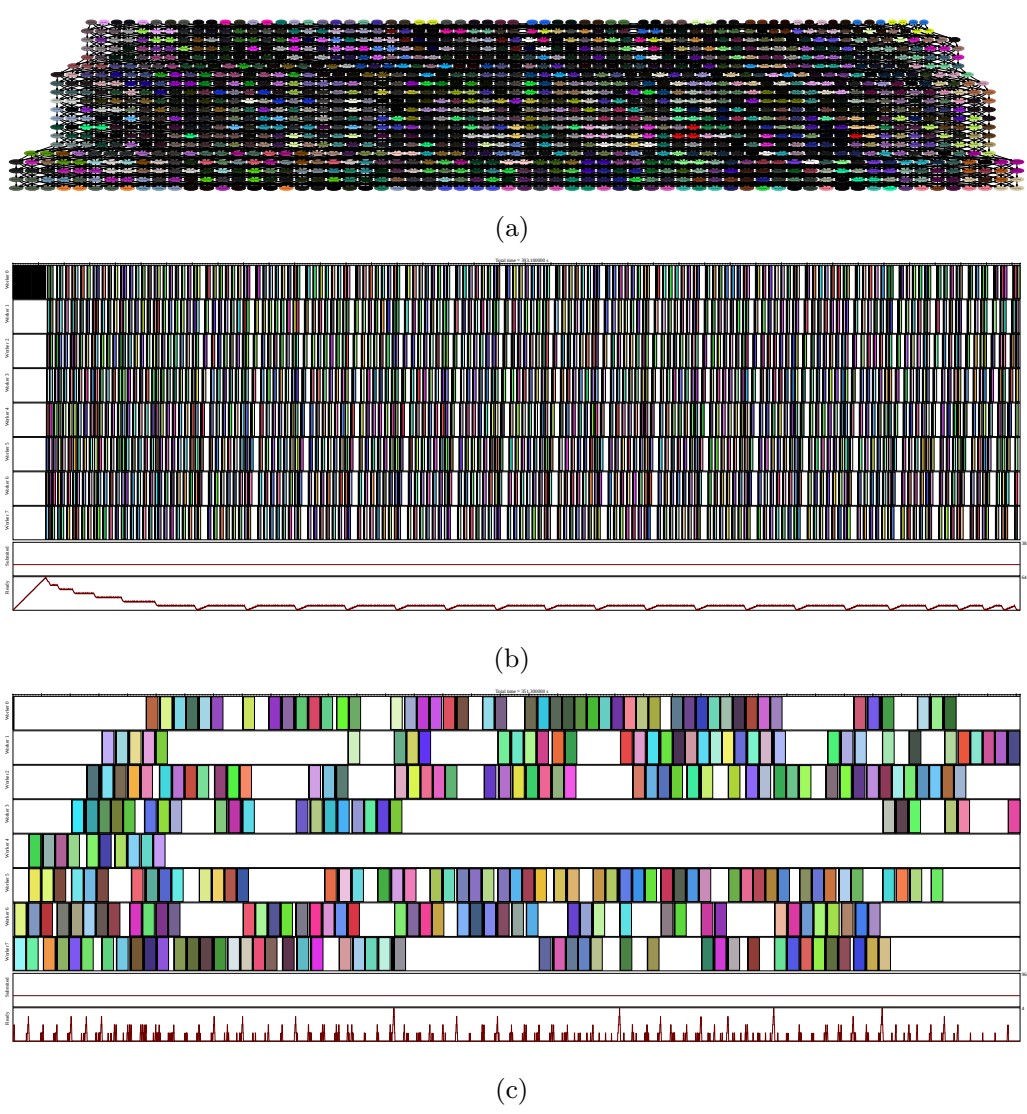

**Figure A3  Example of the Polybench Jacobi 2D clustering and execution.** The graph was generated with parameter $iter = 10$ and $N = 10$. The execution time obtained with granularity, (C), is slower than without granularity, (B), because the overhead is limited and the dependencies makes it difficult to find a meta-graph where the parallelism is not constraint. (A) Clustered graph with $M = 4$. Original graph has 1,280 nodes and estimated degree of parallelism of 56, clustered one has 320 nodes and estimated degree of par-allelism of 4.2. (B) Emulation of the execution of the original graph with 8 threads in 391 units of time. Each original task has a cost of 1, the overhead are 0 per task, 0.1 per push and 0.2 per pop. (C) Emulation of the execution of the clustered graph with 8 threads in 351.3 units of time. Each original task has a cost of 1, the overhead are 0 per task, 0.1 per push and 0.2 per pop.

### Funding

The authors received no funding for this work.

## Competing Interests

The authors declare there are no competing interests.

## Author Contributions

- Bérenger Bramas conceived and designed the experiments, performed the experiments, analyzed the data, contributed reagents/materials/analysis tools, prepared figures and/or tables, performed the computation work, authored or reviewed drafts of the paper, approved the final draft.
- Alain Ketterlin analyzed the data, authored or reviewed drafts of the paper, approved the final draft.

## Data Availability

An implementation of our method is publicly available at https://gitlab.inria.fr/bramas/dagpar. The graphs used in this project are available at https://figshare.com/projects/Dagpar/71198.

## Supplemental Information

Supplemental information for this article can be found online at http://dx.doi.org/10.7717/peerj-cs.247#supplemental-information.

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
