# Peer review of "Improving parallel executions by increasing task granularity in task-based runtime systems using acyclic DAG clustering"

_PeerJ Computer Science, doi:10.7717/peerj-cs.247_

## Round 0.1 · original submission · Major Revisions

The authors improved parallel executions by increasing task granularity in task-based runtime systems using acyclic DAG clustering. The work is sufficient. The organization and writing of the paper are good. For the consideration of potential publication, the authors are suggested further improve the manuscript according to reviewers’ comments, especially pay attention to:

(1) Adding more relevant and necessary references
(2) Giving more accurate statements (please find the second reviewer’s related comments)
(3) Presenting more analysis and discussion on the results.

·

Basic reporting

Problem is well defined and understood, references to other articles are relevant. Structure is conform to professional article.

Experimental design

This work is one step further in study of "acyclic DAG clustering" which permit better performance in case of fine-grain parallelism. To prove validity of this work, they use both simulated and real execution and compare tendencies, which is a technique used in numerous articles. Multiple type of graphs have been used for simulating a wide range of problems.

Validity of the findings

The new clustering algorithm is an enhancement of clustering algorithm describe in another article, additions are well identify and explanations of them are correct.
There is a clear advantage for this new algorithm, this is the key of this article. Results seem reliable and clear to look at.
Unfortunately, simulated results don't match real execution results but it's not a big deal because it give us a good approximation of algorithm efficiency. Authors don't hide this fact and try to explain it.

Additional comments

Good job for finding these new heuristics !

Reviewer 2 ·

Basic reporting

This paper describes a strategy to partition a task graph, with the aim to increase task granularity and speed up the parallel execution time of applications.

The authors provide a good overview of the research topic, and the manuscript and sound and clear. However, I have some major concerns which are detailed in the following comments.

MAJOR REMARKS

Lines 43-45: "When the granularity is too small, the overhead of task management, and the potential data movements, becomes dominant and can dramatically increase the execution time due to the use of synchronizations."
I suggest to include a reference to this work:
Tagliavini, Giuseppe, Daniele Cesarini, and Andrea Marongiu. "Unleashing fine-grained parallelism on embedded many-core accelerators with lightweight OpenMP tasking." IEEE Transactions on Parallel and Distributed Systems 29, no. 9 (2018): 2150-2163.

Line 57: "Hence, the usual graph partitioning methods do not work because they do not take into account the direction of the edges."
The expression "usual graph partitioning methods" is not self-explanatory, the authors should provide at least a reference. For instance:
Hendrickson, Bruce, and Tamara G. Kolda. "Graph partitioning models for parallel computing." Parallel computing 26, no. 12 (2000): 1519-1534.

Line 64: "We provide two variants of the GDCA".
For the sake of clarity, the authors should include a quick description of the two variants in the introduction.

Line 68/69: "DGCA and our approach"
It is not clear if the authors provide a comparison with a "vanilla" DGCA and their variants, or not. Please provide more details or rephrase this sentence.

Line 149: "Implicitly, the hardware or execution model should have some overheads, which could come from the management of the tasks for example, or the minimal execution time will always be obtained without clustering, i.e. any clustering of tasks will reduce the degree of parallelism without offering any advantages."
The idea is clear, but I disagree on the premise: real hardware platforms always suffer for overheads.

Line 170: "However, looking at the edge-cut to measure the quality is not relevant when the final and only objective is the parallel efficiency"
In this context, the adoption of the term "quality" is not clear.

Lines 224-240:
Considering the limited description provided in this paragraph, the contributions of this paper seem to be quite limited. The authors should provide more details to give strength to their contributions.

Lines 242-273:
This section looks disjoint by the rest of the paper, and it also lacks references to the SoA. I suggest a citation to the following paper:
Kestor, Gokcen, Roberto Gioiosa, and Daniel Chavarrıa-Miranda. "Prometheus: scalable and accurate emulation of task-based applications on many-core systems." In 2015 IEEE International Symposium on Performance Analysis of Systems and Software (ISPASS), pp. 308-317. IEEE, 2015.

MINOR REMARKS
The authors should double-check the manuscript for typos, e.g.::
- Line 61: "we use generic algorithm" -> "we use a generic algorithm"
- Line 193: "This strategy ensure" -> "This strategy ensures"
- Line 127: "While this targets the same problem, we arge that execution time is the only criteria that should evaluated."
This sentence contains at least two typos and a grammatical error.

Experimental design

- Lines 135-137: "The real application we used to assess our method solves the transport equation on unstructured meshes. Task-based implementations to solve the transport equation on a grid (i.e. structured and regular mesh) have already been proposed by Salli Moustafa et al. [10]."
Overall, the experimental section is missing a substantial comparison with the state-of-the-art.
Since this application is a potential baseline for the proposed approach, I think that it would be really interesting from a scientific point of view to provide a comparison in the experimental section.

Validity of the findings

no comment

---

## Round 0.2 · accepted · Accept

I am happy that the authors have carefully revised the manuscripts according to the reviewers’ and my comments. Thus, I decide to accept this manuscript for publication. Note again that, when preparing the final latex files for publication, the high-quality figures are needed.

Reviewer 2 ·

Basic reporting

no comment

Experimental design

no comment

Validity of the findings

Taking into account the additional explanations provided in the rebuttal letter, I gave a more in-depth look at the code base to better understand the experimental setup. Finally, I do not have further doubts about the validity of the findings.

Additional comments

The answers provided by the authors in the rebuttal letter are convincing. From my point of view, the manuscript is now in good shape for publication.